# Medical Students and COVID-19: Knowledge, Preventive Behaviors, and Risk Perception

**DOI:** 10.3390/ijerph18020842

**Published:** 2021-01-19

**Authors:** Mansour Alsoghair, Mohammad Almazyad, Tariq Alburaykan, Abdulrhman Alsultan, Abdulmajeed Alnughaymishi, Sulaiman Almazyad, Meshari Alharbi, Wesam Alkassas, Abdulaziz Almadud, Mohammed Alsuhaibani

**Affiliations:** 1Department of Family and Community Medicine, College of Medicine, Qassim University, Qassim 51452, Saudi Arabia; mansouralsoghair@qumed.edu.sa; 2Department of General Surgery, National Guard Hospital, Eastern Province 34217, Saudi Arabia; malmazyad21@gmail.com; 3Department of Internal Medicine, Buraidah Central Hospital, Qassim 52361, Saudi Arabia; Tariqburaykan@gmail.com; 4Department of Anatomy and Histology, College of Medicine, Qassim University, Qassim 51452, Saudi Arabia; aa.alsultan@qu.edu.sa; 5Department of Family Medicine, Uyun Aljawa General Hospital, Qassim 52347, Saudi Arabia; abdulmajeed.alnughaymishi@gmail.com; 6Medical Intern, Unayzah College of Medicine, Qassim University, Qassim 51452, Saudi Arabia; Salmazyad95@gmail.com (S.A.); Abdulazizbinahmad@outlook.com (A.A.); 7Medical Resident, King Fahad Specialist Hospital, Qassim 52347, Saudi Arabia; Meshari818@gmail.com; 8Medical Intern, College of Medicine, Sulaiman Al Rajhi University, Qassim 52347, Saudi Arabia; wesam.alkassas@gmail.com; 9Department of Pediatrics, College of Medicine, Qassim University, Qassim 51452, Saudi Arabia

**Keywords:** COVID-19, knowledge, behavior, risk perception, medical students, Saudi Arabia

## Abstract

Background: The coronavirus disease (COVID-19) pandemic is an international public health threat. This study aimed to evaluate COVID-19-related knowledge, preventive behaviors, and risk perception among Saudi Arabian medical students and interns. Materials and Methods: This cross-sectional study was conducted among fourth- and fifth-year medical students and interns between June and August 2020 at three colleges of medicine in Qassim Region, Saudi Arabia. A previously validated questionnaire was distributed as an online survey. Results: The total mean knowledge score was 12.5/15 points; 83.9% achieved a high score. The mean score of self-reported preventive behavior was 8.40; 94.1% achieved a high score. The overall mean risk perception score was 5.34/8 points; 31.6% achieved a high score. Conclusion: Medical students assessed in this study displayed sufficient knowledge and preventive behaviors regarding the COVID-19 pandemic and an average level of risk perception. Lower scores by younger medical students suggest that they must improve their COVID-19 knowledge and risk perception, as they are a potential source of health information in their communities.

## 1. Introduction

Coronavirus infections are caused by emerging respiratory viruses that are known to cause diseases ranging from the common cold to illnesses involving severe respiratory distress [1]. In 2002, the first pandemic outbreak of the coronavirus that causes severe acute respiratory syndrome (SARS) was associated with a mortality rate of approximately 10%. In 2012, the outbreak caused by Middle East respiratory syndrome coronavirus (MERS-CoV) was associated with a mortality rate of 30–40% [1]. In December 2019, a new outbreak of viral pneumonia of an unknown etiology developed in Wuhan, China. The genetic analysis of the underlying pathogen revealed an enveloped positive-strand ribonucleic acid (RNA) virus belonging to the family Coronaviridae and the order Nidovirales [2]. In February 2020, the World Health Organization (WHO) named the epidemic disease caused by this virus, coronavirus disease (COVID-19) [1]. In response to this serious situation, the WHO declared COVID-19 a public health emergency of international concern [1,2]. COVID-19 infections spread very quickly. By the first weeks of March 2020, many new cases had been reported globally, and COVID-19 was declared a pandemic. In the same month, more than 125,000 cases of COVID-19 were reported in about 118 countries, with more than 4600 deaths [1,2].

The disease is highly infectious, and further studies identified that the most important route of transmission to humans occurred via respiratory droplets or direct contact, with an incubation period ranging from 2 to 14 days [3]. Clinical data have shown the overall mortality rate of COVID-19 infection to be 2.3% in China, which is much lower than that of SARS (9.5%) or MERS (34.4%) [3]. To date, no specific antiviral treatment or vaccine has been confirmed to be effective for treating COVID-19. Therefore, many preventive measures have been identified to help control its transmission [4].

Healthcare providers are the primary individuals in contact with patients who are the main source of infections; thus, they are at high risk of becoming infected themselves. At the end of January, the WHO and the US Centers for Disease Control and Prevention (CDC) published recommendations to help prevent the rapid spread of COVID-19 among healthcare workers [4,5]. Several online training sessions and materials were made available in several languages by the WHO to provide information about COVID-19 prevention strategies and to increase the awareness of healthcare workers treating patients [6].

This knowledge and awareness of the COVID-19 pandemic can influence the risk perceptions of healthcare workers and their ability to engage in preventive strategies. However, some studies have noted a significant gap in information sources available to healthcare workers and have identified low knowledge levels concerning COVID-19 [1]. For example, Pranav et al. concluded that there was a need for a greater number of educational and training programs to reduce the risk of infectious transmission among healthcare students and professionals, and to provide optimal care for patients [7].

Despite numerous studies that have addressed this topic worldwide, there are limited data about the levels of knowledge and behaviors pertaining to COVID-19 treatment in Saudi Arabia. We hypothesized that only a little information is available on COVID-19 and that medical students exercise some incorrect preventive behaviors. Medical students are a source of information for their families and communities; therefore, our study aimed to evaluate COVID-19-related knowledge, preventive behaviors, and risk perceptions among fourth- and fifth-year medical students and interns.

## 2. Materials and Methods

### 2.1. Study Design, Population, and Sampling Methods

This cross-sectional, descriptive study was conducted between 1 June 2020, and 30 August 2020. The study population included both male and female fourth- and fifth-year medical students and interns in the multi-colleges of medicine of Al-Qassim Region, Kingdom of Saudi Arabia (KSA), including the College of Medicine at Qassim University in Buraidah, the College of Medicine at Qassim University in Unaizah, and the College of Medicine at Sulaiman Al Rajhi University. The total number of fourth- and fifth-year students and interns in each university was as follows: College of Medicine at Buraidah (333 students), College of Medicine at Unaizah (221 students), and the College of Medicine at Sulaiman Al Rajhi University (151 students, no female students in the fifth year and internship), and the ratio of male to female students was 2:1. This study was approved by the Committee of Bioethics of the Qassim University, Saudi Arabia (IRB approval # 191406).

A previously validated questionnaire was administered as an online survey that was distributed to all clinical year students through WhatsApp application [2]. The sampling technique followed was the non-randomized convenience method. The survey was expected to take 7 min to complete, and all of the students were reminded to complete the questionnaire within two weeks. The estimated population was 710 students based on information obtained from the student affairs offices.

### 2.2. Study Variables

To assess the subjects’ levels of knowledge, preventive behaviors, and risk perceptions, the questionnaire was divided into four parts. The first part assessed demographic information (age, sex, academic year, university attended, COVID-19 education received, and the source of education), followed by 26 questions to quantify their knowledge, preventive behaviors, and risk perception. Knowledge was assessed through questions about the nature of COVID-19 transmission, its history, symptoms, incubation period, diagnostic procedures, preventative measures, and treatment options.

### 2.3. Scoring Criteria

Medical students’ general knowledge regarding COVID-19 was assessed using 15 questions; the correct response to each question was coded as a “1,” and an incorrect response was coded as a “0.” The total score for knowledge was calculated as the sum of the responses for the 15 questions, which yields a score ranging from 0 to 15. A higher score was indicative of a higher level of knowledge regarding COVID-19. Using 50% and 75% as cutoff points, participants were classified as having low knowledge if their scores ranged between 1 and 7, scores ranging between 8 and 11 indicated average knowledge, and scores ranging between 12 and 15 indicated a high level of knowledge.

The self-reported preventive behaviors were assessed by binary responses to nine questions; the “yes” response was coded as “1,” and the “no” response was coded as “0.” The total score for the self-reported preventive behaviors was calculated as the sum of the responses to the nine questions, which yielded a score ranging between 0 and 9; a greater degree of self-reported preventive behaviors was indicated by a higher score. Using 50% and 75% as cutoff points, medical students were considered to have a low level of preventive behavior if their scores ranged between 1 and 4, an average level for scores ranging between 5 and 6, and a high level for scores ranging between 7 and 9.

Risk perception was assessed by two questions whose responses were recorded on a 4-point Likert scale, with categories ranging from “strongly disagree” (coded as “1”) to “strongly agree” (coded as “4”). The total risk perception score was calculated as the sum of the responses to the two questions, generating a score ranging from 2 to 8. A higher score was indicative of a greater level of perception pertaining to the risk of COVID-19. Using 50% and 75% as cutoff points, medical students were classified as having low risk perception for scores ranging from 1 to 4, scores ranging from 5 to 6 were considered an average level of perception, and scores ranging from 7 to 8 were considered to be a high level of perception.

### 2.4. Statistical Analysis

Data are presented as counts, proportions (%), ranges, means, and standard deviations whenever appropriate. The comparisons of knowledge, behavior, and risk perception among the basic demographic data of medical students were conducted using the Mann Whitney U test (in cases of two categories) or the Kruskal Wallis test (in cases of three categories). *p*-values < 0.05 were considered to be statistically significant. Normality, statistical interactions, and collinearity (i.e., the variance inflation factor) were also assessed with the Kolmogorov-Smirnov and Shapiro-Wilk tests, and *p*-values < 0.05 were considered indicative of skewed data distributions. Correlation analyses were also conducted to assess the linear relationships between knowledge, behavior, and risk perception. All data analyses were performed using Statistical Packages for Software Sciences (SPSS) version 21 software (IBM Corporation, Armonk, New York, NY, USA).

## 3. Results

Of the 710 medical trainees approached, 323 responded (response rate 45.5%); their basic demographic data are presented in Table 1. The most commonly listed source of COVID-19 information was material provided by the local Ministry of Health (75.7%), followed by the CDC (68.9%) and the WHO (63.9%). As shown in Figure 1, published articles were the least frequently listed sources of information (43.4%).

### 3.1. Assessment of Knowledge of COVID-19

In the assessment of medical students’ knowledge regarding COVID-19 (Table 2), respondents exhibited a high level of knowledge for nearly all statements, except for item #12 for which incorrect responses were common. The top five statements that medical students responded to correctly were the following: “The first case of COVID-19 was identified in Wuhan, China” (98.8%); “It is transmitted through respiratory droplets such as through coughing and sneezing” (97.5%); “COVID-19 is a respiratory infection caused by a new species of virus of the coronavirus family” (96.6%); “The disease can be prevented by avoiding close contact such as handshakes and kissing, not attending in-person meetings, and by frequently disinfecting the hands” (96.6%); and “A medical mask is useful to prevent the spread of respiratory droplets through coughing” (96%). Most participants incorrectly identified the correct answer to the statement, “All people in the society should wear face masks when going outside” (6.2%).

### 3.2. Self-Reported Assessment of Preventive Behaviors

Almost all medical students (varying from 88.5% to 96.3%) self-reported good preventive behaviors toward COVID-19, as shown in Table 3.

### 3.3. Assessment of Medical Students’ Risk Perception regarding COVID-19

Of the respondents, 38.1% disagreed that they may be infected with COVID-19 more easily than others, while 28.8% agreed. We also observed that nearly half of the medical students (47.7%) agreed that they were afraid of being infected with COVID-19, whereas 26.9% stated otherwise. The level of knowledge, self-reported preventive behavior, and risk perception among medical students are summarized in Table 4.

### 3.4. Assessment of the Correlations between Knowledge, Preventive Behavior, and Risk Perception Scores

Knowledge and self-reported preventive behavior scores were weakly positively correlated, and this relationship was highly statistically significant (r = 0.193; *p* < 0.001), suggesting that when knowledge increased, self-reported preventive behavior was also likely to increase. The correlation between knowledge and risk perception was not statistically significant (r = 0.042; *p* = 0.447). There was, however, a weak positive correlation between self-reported preventive behavior and risk perception scores (r = 0.119; *p* = 0.032), indicating that a higher self-reported preventive behavior score was associated with an increase in risk perception.

When comparing the knowledge, self-reported preventive behavior, and risk perception score among medical students and interns, we noted that medical students studying at Almleda Qassim University had significantly better risk perception score than those studying at other universities (*p* = 0.001). Furthermore, fourth-year students had significantly less knowledge (*p* = 0.009) and a lower perception score (*p* = 0.001) than other academic year students. However, age, sex, and having medical knowledge about COVID-19 were not significant when compared to knowledge, self-reported preventive behavior, and risk perception scores (All *p* > 0.05).

## 4. Discussion

Senior medical students, as well as interns, are on the frontline in the fight against the novel coronavirus, and they have always been at risk of contracting the infectious disease. For these reasons, it is necessary to evaluate their knowledge, assess their preventive behaviors, and measure their risk perception in regards to the COVID-19 pandemic. The outcomes of this research could be useful for health policymakers and medical educators to design a systematic plan to ensure that medical students are aware of the COVID-19 pandemic and the necessary preventive measures. A literature search conducted in Saudi Arabia revealed limited available data from studies that measured medical students’ knowledge, preventive behaviors, and risk perception during the COVID-19 pandemic. In this study, nearly all students (83.9%) exhibited a high level of knowledge regarding COVID-19; the rest exhibited either an average (14.6%) or a low level (1.5%) of knowledge. The overall mean knowledge score was 12.5 (SD 1.47). Consistently, the current literature indicates that many people, including healthcare workers, medical students, and the general public, demonstrated a high level of knowledge regarding the pandemic, varying from 69% to 91% [2,7,8,9,10,11,12,13,14]. On the other hand, during the MERS viral outbreak, studies assessing groups’ knowledge of the outbreak reported inconsistent results. For instance, in Saudi Arabia, Khan et al. [15] documented that healthcare workers’ knowledge of MERS was “good” among 73.2% of respondents, which was in line with our own reports, but contrary to those of Nuor et al. [16] in which 67.6% of healthcare providers exhibited a “poor” level of knowledge. Furthermore, several studies have indicated that age was a factor associated with knowledge about the virus [1,8,10,16], as increased age was associated with an increased level of knowledge. This was not observed, however, in our study; although the results showed that the older age group (≥25 years) exhibited better knowledge, this did not reach statistical significance. Similarly, reports by Al Hanawi et al. [8] as well as Khan et al. [15] indicated that sex was a factor associated with knowledge; however, both studies reported disparate results. In our study, there was no significant relationship was observed between sex and the level of COVID-19 knowledge (*p* > 0.05), which contradicts previous findings. We believe that the source of information (curriculum and extra-curriculum) for both sex and age were similar during the pandemic. Additionally, the number of students who had no medical knowledge (as per the student self-assessment) was extremely low (4.3%).

Moreover, it is important to note that academic level was the only significant factor associated with the level of COVID-19 knowledge, as the 4th-year students exhibited significantly lower knowledge scores than those at the 5th-year level or interns. This is comparable to the results of other studies conducted in Saudi Arabia [8], Egypt [10], and Uganda [11], where level of education was a factor that was significantly associated with the level of knowledge. However, a study conducted in Iran [2] found no significant difference between the level of education and knowledge, which is contrary to our results. We believe that academic level and exposure to clinical practice would influence the level of students’ knowledge as an intern, and fifth-year students gain more experience in clinical training.

The data revealed that medical students exhibited an overwhelmingly high degree of preventative behaviors to protect against COVID-19 (94.1%); only a few students exhibited average (5.3%) and low-level preventive behaviors (0.6%). This finding mirrored that of a study conducted by Taghir et al., [2], who found that 94.2% of medical students exhibited a high degree of self-reported preventive behaviors in response to the pandemic. Moreover, our data showed that no sociodemographic variables were significantly associated with self-reported preventive behavior scores, which was also similar to the findings of Taghir et al. [2].

Apart from measuring the knowledge and behavior of medical students related to the COVID-19 pandemic, we also investigated their levels of risk perception, which was at an average level among medical students. This is also consistent with the average level of risk perception that was reported by Taghir et al. [2]. In contrast, healthcare workers, as well as medical and allied health science students, exhibited positive perceptions about the pandemic, as reported by researchers in the United Arab Emirates and in India [1,9]. Furthermore, we also learned that the academic level was another factor that influenced risk perception, as the 4th-year students had significantly lower scores than the other groups. The level of education was also a factor known to be associated with risk perception, as reported by studies conducted in Iran [2] and Korea [12]. On the other hand, according to Bhagavathula et al. [1], age and profession were significant factors associated with poor risk perception. However, in our study, age was not significantly associated with risk perception, though we could not assess whether profession was an associated factor since our subjects were all undergraduate medical students. Moreover, we believe that the time when the pandemic started, cultural aspects, and trust of health authorities might influence knowledge, behavior, and risk perception of individuals in different countries.

One of the highlights of our results was that there was a positive, significant correlation between the knowledge and self-reported preventive behavior scores, suggesting that as medical students’ knowledge of COVID-19 increased, so did their self-reported preventive behaviors. On the other hand, there was no significant correlation between knowledge and risk perception scores. These results were not consistent with those of Kim and Choi [12], who reported a significant correlation between the levels of knowledge and preventive behaviors. However, in the study by Taghir et al., [2], they identified a negative correlation between preventive behavior and risk perception, arguing that while self-reported behavior scores increased, risk perception was likely to decrease. In our study, we found a positive correlation between self-reported preventive behaviors and the risk perception score, indicating that as the self-reported behaviors increased, the risk perception also increased, which was in contrast to the results of the Iranian study described above [2].

To learn more about the current pandemic, it is necessary to provide everyone with access to sources of information. In this study, the most frequently noted source of COVID-19 information was material provided by the local Ministry of Health (75.7%), followed by the CDC (68.9%), and the WHO (63.9%). Due to the influence of social media, many people also sought information on the internet. This has also been consistently reported in the literature, where the majority of individuals read information about the pandemic on public health websites [1,9,10,11,13,15,16,17,18]. In our study, 57.6% of medical students reported using social media to obtain information about COVID-19, which was also in line with previous findings.

This study had some limitations. In particular, the study was conducted in only a single region, which may not be representative of all university students in Saudi Arabia. Additionally, the questionnaire was conducted online due to the mitigation measures in place in response to the COVID-19 pandemic, which might have affected the response rate and generalizability among the university students. Nevertheless, this study provides an estimate of medical students’ current COVID-19 knowledge and risk perception in Saudi Arabia during the pandemic.

## 5. Conclusions

Medical students play an important role as volunteers during emergency, including pandemics. This study revealed that current knowledge and self-reported preventive behaviors of medical students pertaining to the COVID-19 pandemic were sufficient, although risk perception was average and could be improved. Medical students studying at the Almleda Qassim University had a better outlook in terms of risk perception. However, younger medical students need to improve their Knowledge of COVID-19 and risk perception since they are a potential source of health information within their communities. Thus, more education must be provided to fill the gaps, especially for those in the earlier academic levels of medical training, as they are a vulnerable group that may underestimate the risks of COVID-19.

## Figures and Tables

**Figure 1 ijerph-18-00842-f001:**
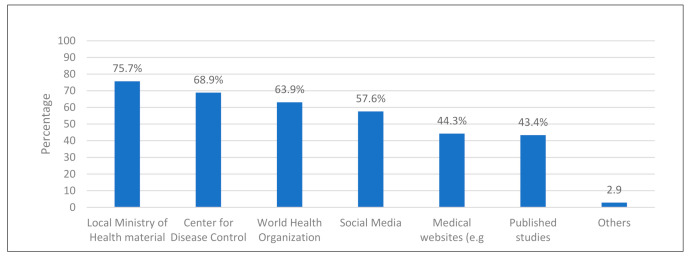
Sources of COVID-19 information.

**Table 1 ijerph-18-00842-t001:** Basic demographic data of medical students (*n* = 323).

Demographic Data	N (%)
Age group	
• <25 years	205 (63.5%)
• ≥25 years	118 (36.5%)
Sex	
• Male	237 (73.4%)
• Female	86 (26.6%)
University attended	
• Unaizah Qassim University	96 (29.7%)
• Almleda Qassim University	116 (35.9%)
• Sulaiman Al Rajhi University	111 (34.4%)
Academic level	
• 4th year	103 (31.9%)
• 5th year	100 (31.0%)
• Intern	120 (37.1%)
Having medical knowledge about COVID-19	
• Yes	309 (95.7%)
• No	14 (4.3%)

**Table 2 ijerph-18-00842-t002:** Assessment of medical students’ knowledge regarding COVID-19 (*n* = 323).

Statement	Correct Answer N (%)
1. COVID-19 is a respiratory infection caused by a new species of virus in the coronavirus family	312 (96.6%)
2. The first case of COVID-19 was diagnosed in Wuhan, China	319 (98.8%)
3. The origin of COVID-19 in humans is likely through transmission from bats	296 (91.6%)
4. Its common symptoms are fever, cough, and shortness of breath, but nausea and diarrhea are reported rarely	279 (86.4%)
5. Its incubation period is up to 14 days, with a mean of 5 days	301 (93.2%)
6. It can be diagnosed by PCR testing of samples collected from nasopharyngeal and oropharyngeal discharge or from sputum and bronchial washing	295 (91.3%)
7. It is transmitted through respiratory droplets such as those generated by coughing and sneezing	315 (97.5%)
8. It is transmitted through close contact with an infected person (especially by family members and in crowded places and healthcare centers).	306 (94.7%)
9. The disease can be prevented through handwashing and personal hygiene	302 (93.5%)
10. A medical mask is useful to prevent the spread of respiratory droplets during coughing	310 (96.0%)
11. The disease can be prevented through maintaining no close contact, such as handshakes and kissing, not attending in-person meetings, and frequently disinfecting the hands	312 (96.6%)
12. All people in the society should wear face masks when going outside	20 (6.2%)
13. Only during aerosol generation procedures such as intubation, suction, bronchoscopy, and cardiopulmonary resuscitation do you have to wear an N95 mask	207 (64.1%)
14. The disease can be treated by usual antiviral drugs	188 (58.2%)
15. If symptoms appear within 14 days from direct contact with a suspected case, the person should inquire at a nearby ministry of health center	278 (86.1%)

**Table 3 ijerph-18-00842-t003:** Assessment of self-reported preventive behavior of medical students in response to COVID-19 (*n* = 323).

Statement	Yes (%)
1. I canceled or postponed meetings with friends, eating out, and sporting events.	293 (90.7%)
2. I reduced the use of public transportation.	297 (92.0%)
3. I went shopping less frequently.	300 (92.9%)
4. I reduced visits to closed spaces, such as the library, theatre, and cinema.	302 (93.5%)
5. I avoided coughing around people as much as possible.	311 (96.3%)
6. I avoided places where a large number of people gathered.	311 (96.3%)
7. I increased the frequency of cleaning and disinfecting items that can be easily touched with my hands (i.e., door handles and surfaces).	286 (88.5%)
8. I washed my hands more often than usual.	304 (94.1%)
9. I discussed COVID-19 preventions with my family and friends.	310 (96.0%)

**Table 4 ijerph-18-00842-t004:** Level of knowledge, self-reported preventive behavior, and risk perception regarding COVID-19 (*n* = 323).

Variables	N (%)
Level of knowledge	
• Low	5 (1.5%)
• Average	47 (14.6%)
• High	271 (83.9%)
Knowledge score (mean ± SD)	12.5 ± 1.47
Level of self-reported preventive behavior	
• Low	2 (0.60%)
• Average	17 (05.3%)
• High	304 (94.1%)
Self-reported preventive behavior score (mean ± SD)	8.40 ± 1.01
Level of risk perception	
• Low	102 (31.6%)
• Average	149 (46.1%)
• High	72 (22.3%)
Risk perception score (mean ± SD)	5.34 ± 1.49

## Data Availability

The data presented in this study are available on request from the corresponding author.

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
