# Peer review of "Medical Students and COVID-19: Knowledge, Preventive Behaviors, and Risk Perception"

_ijerph, 2021, doi:10.3390/ijerph18020842_

Round 1

Reviewer 1 Report

Dear autor and co-authors,

First of all, I would like to congratulate you for the effort in these complicated moments.

The manuscript is very well designed but I have some doubts or suggestions:

  1. This study was approved by the committee of bioethics of the Qassim University, but all studies have a registration code in that ethics committee, which is the code of the ethics committee of this study?
  2. in the Dicussion section (lines 215-216) why do you think it did not achieve statistical significance?

Kinds Regards

Author Response

Dear Reviewer,

Thank you for your comments and suggestions, Please see the attachment.

Reviewer 2 Report

During the outbreak of COVID-19 studies on differnet aspects of the pandemic become interesting. These includes studies with medical personel and their perspective on the disease. The mansucript is the example of such studies as it focuses on medical students' knowledge, risk perception and preventive behaviour. Analyses of such group are important. The study is properly designed and presented. Although I would suggest the Authors to consider some changes.

  1. In the description of the questionnaire and scoring methods is stated that the range of scores is from 1 to 8, etc, what implicates that none incorrect answer (which is scored 0) is given, while it is plausible that at least one respondent might give all incorrect answers. Thus the scores should be: from 0 to 8, etc. Table 1 indicates two types of knowledge on COVID-19 - yes or no - while the scoring method indicate 3 levels of knowledge. This should be  clarified.
  2. As no all students have replied to the questionnaire some information how those who participated might be different from the rest should be given.
  3. The final sample includes 4th and 5th year medical students and interns, thus persons with different chance for clinical contacts/experiences with COVID-19 patients and/or real contact. This aspect was neither included in the analyses nor in the discussion.  I would suggest some comments on this  issue.

Author Response

(The authors gave the same response as above.)

Reviewer 3 Report

A well written evaluative study of medical students' knowledge, preventive behaviours and risk perception of medical students and interns in Saudi Arabia.  The survey is based on a previously validated questionnaire from Iran (reference 2).   With regards to the distribution of the survey 86 and 87, who was the sender of the online questionnaire via whatsapp?  Did the students also get a reminder to complete the survey and also a set deadline?  This may have in influence on the response rate.  

It would be useful to see an example of questions in the supplementary material if the journal permits.  Other than this the methodology regarding scoring criteria, and how the scores were translated to ordinal categories were clear.

Results

The findings and demographics are transparent, and it can be seen that the ratio of male to female is 3:1.  The distributions of the responses for each university is roughly proportionate.  

Lines 159-164, If the author could re-order the top five statements that the medical students responded to to start with the statement with the highest percentage, e.g. "The first case........ (98.8%); ............(97.5%); ............96.6%; ...........96.6% and .........96%".

In terms of 164-165, the question is a bit problematic, as I agree that all people should wear face masks when going outside. However, there are certain groups that may not be able to wear face masks, such as babies and toddlers, of which they would need to be kept safe some other way.  

Discussion

Line 199-200

"The outcomes of this research..........."

I would say that it would help health policy makers/medical educators plan education and training programmes to ensure that medical students are aware of the pandemic and what to do in the case of COVID-19....

Conclusion

I agree with the conclusion that health care professionals such as doctors need to be educated and receive training to deal with novel virus and pandemics such as COVID-19

Author Response

(The authors gave the same response as above.)

Reviewer 4 Report

The study starts from an interesting theme, the perception of risk in medical students and their role in spreading a culture of prevention. which has some other specific comparison searches. The literature considered is specific but taking into consideration very different countries it does not adequately take into account the influence of cultural factors and of different timing and incidence of spread of the pandemic in the various places, in order to be able to make a direct comparison.
The methodology applied and above all the statistical processing is decidedly basic, as it is limited to response percentages. No comparisons are made with more elaborate statistics with respect to the variables taken into consideration (age, gender, type of university) to understand their influence on attitudes and perception of risk.
The conclusions therefore seem little supported by solid and significant results.

Author Response

(The authors gave the same response as above.)
